# Adverse Childhood Experiences and Oral Health Outcomes in U.S. Children and Adolescents: A Cross-Sectional Study of the 2016 National Survey of Children’s Health

**DOI:** 10.3390/ijerph182312313

**Published:** 2021-11-23

**Authors:** Alyssa Simon, Jamie Cage, Aderonke A. Akinkugbe

**Affiliations:** 1Division of Epidemiology, Department of Family Medicine and Population Health, School of Medicine, Virginia Commonwealth University, Richmond, VA 23284, USA; simona8@vcu.edu; 2School of Social Work, Virginia Commonwealth University, Richmond, VA 23284, USA; jlcage@vcu.edu; 3Institute for Inclusion, Inquiry and Innovation, Virginia Commonwealth University, Richmond, VA 23284, USA; 4Department of Dental Public Health and Policy, School of Dentistry, Virginia Commonwealth University, Richmond, VA 23284, USA

**Keywords:** adverse childhood experiences, violence, oral health, pediatric dentistry, mental disorders, caregivers

## Abstract

This study investigated the cross-sectional associations between exposure to nine Adverse Childhood Experiences (ACEs) and U.S. children’s and adolescent’s oral health outcomes. Data from 41,294 participants of the 2016 National Survey of Children’s Health (NSCH) were analyzed. Past year exposure to ACE, oral health outcomes (decayed teeth, bleeding gums, and condition of the teeth), and child and caregiver sociodemographic factors were self-reported. Using SAS v. 9.4, propensity score weighted, multilevel survey-logistic regression estimated adjusted odds ratios (AORs) and 95% Confidence Intervals (CIs) of the proposed associations. The overall mean (SE) age was 8.9 (0.1) years with 51% being male. Fifty-four percent (54%) identified as non-Hispanic white, and 12% as non-Hispanic black. The prevalence of the nine ACE measures ranged from 3% for caregiver death to 25% for financial hardship and parental divorce. Children who experienced caregiver mental illness, when compared to those who did not, were more likely to report decayed teeth (AOR: 1.73 (95% CI: 1.24, 2.42)) and the condition of their teeth as fair/poor (AOR: 1.60, 95% CI: 0.61, 4.19). Children in households with financial hardship were about twice as likely to report dental caries (AOR: 1.85, 95% CI: 1.50, 2.29) and have fair/poor teeth (AOR: 1.87, 95% CI: 1.40, 2.51) and bleeding gums (AOR: 2.39, 95% CI: 1.48, 3.86). ACEs appear to be associated with worse oral health outcomes among children and adolescents. Nevertheless, the cross-sectional nature of this study precludes a causal interpretation of these findings and necessitates more research to elucidate the oral health impacts of exposure to ACEs in longitudinal follow-up studies.

## 1. Introduction

The 2000 U.S. Surgeon General’s report on oral health labeled oral diseases a “silent epidemic” that affects all age groups [1]. Oral disease conditions include dental caries, severe gum disease and tooth loss, with the most common being dental caries [1,2]. About one in five children between the ages of 5 to 11 years and one in seven adolescents between the ages of 12 and 19 years have at least one untreated tooth decay [3,4]. Poor oral health in children poses unique challenges. Indeed, dental pain is a distinguishing factor. In addition to this, children with poor oral health, including dental caries, are significantly likely to miss and to underperform in school [5], use emergency departments for pain management, and are more likely to have delays in growth and development [6,7,8]. Furthermore, poor oral health among young children puts them on a trajectory for poor oral health in later childhood [9,10]. Kragt et al., 2016, reported that children who experienced severe dental caries at six years old were at significantly higher odds of a poor oral health-related quality of life (OHRQoL) at age ten [9]. The negative impacts of dental caries on quality of life extends beyond the affected child to their families [11]. Notably, specific groups of children disproportionately experience poor oral health. Mexican American and non-Hispanic black children are more likely to experience tooth decay than their non-Hispanic white counterparts are [3,12]. Furthermore, children from low-income households are twice as likely to have tooth decay as are children from higher-income households [3,12].

The determinants of poor oral health outcomes include several environmental, behavioral and lifestyle-related risk factors [13]. Recent studies have also focused on upstream or distal etiological factors, among which are parental psychosocial factors. Maternal psychosocial and behavioral factors affect a child’s risk for many childhood conditions including poor oral health [14,15]. Moreover, adverse childhood experiences (ACEs), defined as negative life events experienced between birth and the age of 18 years [16], negatively affect overall health and wellbeing with significant morbidity and mortality consequences [17]. Although parental divorce, and experiencing economic hardships are the most commonly occurring ACEs in U.S. children [18,19], Brown et al. found that living with a mentally ill household member and experiencing physical abuse were associated with having complex health profiles in a population of child welfare recipients [18].

Few studies have examined exposure to ACE and oral health [20,21,22,23]. Findings from these studies suggest that ACEs, with an emphasis on the ACE of experiencing child abuse, had adverse effects on oral health outcomes [20,21,22,23,24]. Notably, two of these studies were conducted in adults [20,23] and, thus, subject to several biases including misreporting, repression of memories or refusal to disclose sensitive information by adult respondents. For studies among school aged children [21,22,24], ACEs measures were combined into a score, thereby masking the effects of individual ACEs. A previous study [25] examined the effects of individual ACEs on untreated oral healthcare needs in children but more robust oral health data now exist in the 2016 National Survey of Children’s Health (NSCH) for replication.

This study seeks to investigate the effect of exposure to nine ACEs (caregiver mental illness, exposure to neighborhood violence, financial hardship, caregiver divorce, death, incarceration, domestic violence in the home, drug/alcohol abuse, being treated unfairly due to race/ethnicity) on the oral health outcomes (dental caries, bleeding gums) and self-rated condition of the teeth (proxy for oral health related quality of life) in a national sample of U.S. children and adolescents.

## 2. Materials and Methods

### 2.1. Data Source, Study Design and Population

Data for this cross-sectional study come from 41,294 respondents of the 2016 National Survey of Children’s Health (NSCH). The NSCH is a mail and online survey conducted by the data resource center for child and adolescent health aimed at understanding the health issues experienced by U.S. children. Randomly selected addresses from civilian, non-institutionalized households across the U.S. were mailed instructions to access the survey online. After at least one reminder letter, households that had not accessed the online survey were mailed a paper-screening questionnaire. Participants fill out an initial screener with the age and sex of all children in the household. Additional information requested on the four youngest children in the household included their race/ethnicity, English proficiency and the presence of special health care needs. Subsequently, one child from each household was randomly selected to participate in the survey that had 3 versions for the following age groups: 0–5, 6–11 and 12–17 years [26]. The respondents to the NSCH survey were parents or guardians who knew the dependent child or adolescent’s health best.

### 2.2. Exposures

The predictors of interest were caregiver reports of child/adolescent experiences of diverse ACEs in the past 12 months. These ACEs were caregiver mental illness, neighborhood violence, caregiver divorce, caregiver death, domestic violence in the home, parental drugs/alcohol use, caregiver incarceration, household financial hardship, and unfair treatment because of race/ethnicity. Responses were recorded as a binary (Yes or No).

### 2.3. Outcomes

Caregivers responded “Yes” or “No” to the question of study child/adolescent having had decayed teeth or cavities and bleeding gums in the past 12 months. Moreover, they also rated the condition of the study child’s teeth, with response options that we categorized into “Excellent/Very Good/Good” and “Fair/Poor”. We used this measure as a proxy for oral-health-related quality of life (OHRQoL). While there appears to be no single, universally accepted, parameter quantifying OHRQoL, theoretical models including functional limitations, physiological pain thresholds, psychological discomfort and multiple dimensions of disability have been proposed [27,28], but are unavailable in the NSCH data source.

### 2.4. Covariates

*Child specific characteristics*: Age—modeled as continuous and categorized as 0–5, 6–11 and 12–17 years for descriptive purposes, gender (male, female), race/ethnicity (Hispanic/Latino, non-Hispanic black, non-Hispanic white, non-Hispanic other). Whether the child was born in the United States (yes, no), presence of childhood anxiety (yes, no), depression (yes, no) and special health care needs (yes, no).

*Caregiver characteristics*: Gender (male, female), maternal age at birth of the study child and education (≤high school, some college, ≥bachelor’s degree).

*Household/family characteristics* include primary language spoken in the home (English, Spanish, other), family poverty to income ratio and family structure (two parents (married or unmarried), single parent and other) and number of children in the household.

### 2.5. Statistical Analysis

Data analysis was restricted to participants with no missing data on all variables, including the outcomes, exposures and covariates. Furthermore, 1018 0–5-year-olds with no teeth were excluded, given that it was not possible for them to have dental diseases in the absence of teeth. This resulted in an effective final sample size of 41,294. Data analysis began with summary statistics for the child, caregiver and household/family characteristics. Next, descriptive statistics of frequencies and weighted percentages were reported for the individual ACE measures. We created separate propensity scores for each of the 9 ACE measures using SAS’ PROC PS MATCH (SAS Institute, Cary, NC, USA) that was adjusted for each of the child, caregiver and household/family characteristics. The corresponding inverse probability of exposure weight (IPEW) was multiplied with the survey sampling weight for the NSCH study to form the final adjustment weight. The final model was a weighted (IPEW *survey sampling weight) multilevel survey–logistic regression that estimated the odds ratios and 95% confidence intervals (CIs) of the independent associations between exposure to each of the 9 ACEs and the respective oral health outcomes. Data analyses were conducted in SAS v. 9.4 (SAS Institute, Cary, NC, USA) and accounted for the complex survey and sampling design of the NSCH with degrees of freedom calculated using SAS. Reporting for this study adhered to the Strengthening the Reporting of Observational studies in Epidemiology (STROBE) guidelines [29].

## 3. Results

### 3.1. Child Characteristics

The mean age of the study participants was 8.9 years (SE: 0.1), slightly less than half (49%) were female and 20% had a special healthcare need. The majority (54%) were non-Hispanic white, 23% were Hispanic/Latino, and 12% non-Hispanic black (Table 1). Overall, 5% had the condition of their teeth rated as fair or poor, 2% had bleeding gums and 11% experienced tooth decay in the past year (Table 1). The prevalence of experiencing the nine ACE measures in the past year ranged from 3%, for caregiver death, to 25%, for financial hardship and parental divorce. Eight percent experienced caregiver mental illness, and 4% experienced unfair treatment because of their race/ethnicity (Table 2).

### 3.2. Caregiver Characteristics

The caregiver respondents were more likely to be female (70%), 45% had a bachelor’s degree or higher, with a mean (SE) maternal age at the birth of the study child of 29 (0.1) years (Table 1). Of the male caregivers, 6% reported that the child experienced caregiver mental illness, compared to 9% of the female caregivers. Furthermore, 2% of the male caregivers as compared to 4% of the female caregivers reported that the child experienced neighborhood violence (Appendix A Table A1).

### 3.3. Household Characteristics

Respondents reported a variety of family structure types: 68% reported a currently married two-parent household, 9% reported a two-parent household that were not currently married, 15% reported a single-parent (mother only) household structure, and 8% reported another family structure (currently, formerly, or never married; or no parent in the household) (Table 1). Five percent of the currently married two-parent households and 11% of their unmarried counterparts reported a child as having experienced caregiver mental illness. Likewise, 14% of the single-parent households and 17% of “other” households reported that the child had experienced caregiver mental illness. Similarly, reports of a child having experienced neighborhood violence was lowest in two-parent, currently married households at 2% as compared to 4% in two-parent, unmarried households, 8% in single-parent households and 9% in “other” households (Appendix A Table A1). Most children were from households with two or three children (62%) that are predominantly English speaking (87%)—Table 1.

### 3.4. ACEs and Dental Caries

Caregiver mental illness was associated with greater unadjusted and adjusted odds of tooth decay (OR: 2.10; 95% CI: 1.67, 2.65 and AOR: 1.73; 95% CI: 1.24, 2.42), respectively). Similarly, all the nine ACE measures were associated with statistically significant greater unadjusted odds of dental caries. However, the odds ratio for exposure to financial hardship (AOR: 1.85, 95% CI: 1.50, 2.29), caregiver divorce (AOR: 1.87, 95% CI: 1.28, 2.71), neighborhood violence (AOR: 2.09, 95% CI: 1.48, 2.95) and drug and alcohol problems (AOR: 2.11, 95% CI: 1.35, 3.31) remained statistically significant upon covariate adjustment (Figure 1 and Table 3).

### 3.5. ACEs and Bleeding Gums

Independent exposure to all nine ACE measures were associated with bleeding gums in the unadjusted analysis. For instance, caregiver mental illness and financial hardship were associated with an unadjusted odds ratio (95% CI) of bleeding gums of 1.71 (1.15, 2.53) and 3.23 (2.21, 4.71), respectively. Upon a covariate adjustment, the odds ratios for financial hardship (AOR: 2.39, 95% CI: 1.48, 3.86), caregiver mental illness (AOR: 1.75, 95% CI: 1.06, 2.90), neighborhood violence (AOR: 2.97, 95% CI: 1.54, 5.72), domestic violence (AOR: 3.54, 95% CI: 1.28, 9.77) and drugs/alcohol problems (AOR: 2.66, 95% CI: 1.42, 5.00) remained statistically significant (Figure 1 and Table 3).

### 3.6. ACEs and Fair/Poor Condition of Teeth

Independent exposure to any of the nine ACE measures was associated with reports of a fair/poor condition of the teeth in the unadjusted analysis. For instance, caregiver mental illness and financial hardship were associated with greater unadjusted odds (95% CI) of fair/poor condition of the teeth of 2.43 (1.73, 3.41) and 3.18 (2.48, 4.08), respectively. Upon covariate adjustment, the odds ratios for neighborhood violence (AOR: 2.56, 95% CI: 1.62, 4.06), financial hardship (AOR: 1.87, 95% CI: 1.40, 2.51), drug and alcohol problems (AOR: 3.98, 95% CI: 1.33, 11.8) and being treated unfairly due to race/ethnicity (AOR: 3.77, 95% CI: 1.34, 10.6) remained statistically significant (Figure 1 and Table 3).

## 4. Discussion

The current study aimed to expand knowledge on the association between ACEs and oral health. The findings indicate that exposure to diverse adverse childhood experiences in the past year were associated with having worse oral health outcomes, as measured by dental caries, self-rated condition of the teeth (proxy for oral health related quality of life (OHRQoL)) and bleeding gums. The most frequently occurring ACEs were financial hardship and parental divorce, which were experienced by 25% of the children, a finding similar to previous reports [18,19,20].

Bleeding gums is a consequence of inadequate attention (including a lack of supervision) paid to tooth brushing either by the child/adolescent or by the caregiver of the younger children. Indeed, bleeding gums is a better indicator of poor oral healthcare practices that is readily visible than dental hard tissue consequences such as tooth decay. Furthermore, poor oral health among children and adolescents may be reflective of chronic caregiver stress brought on by economic hardship that manifests in the form of inadequate parenting, caretaking and neglect [30], and less attention given to preventive oral healthcare needs [14]. Structural and psychosocial factors, especially poverty and financial hardships, may also contribute to parental/caregiver strain that results in the parent’s/caregiver’s inability to provide consistent dental care and oversight. Of note, we avoid using the terminology “dental neglect” when discussing the potential lack of dental care and oversight because dental neglect assumes “willful failure of parent or guardian, despite adequate access to care… [31]”. Considering that 25% of our sample reported experiencing financial hardships, it is likely that they did not have access to adequate dental care. Thus, our findings are contextualized within this frame of reference. Additionally poverty and financial hardship are associated with lower rates of private dental coverage [32] and dental coverage for individuals with Medicaid [33], which, in turn, has been linked to lower rates of preventative dental visits and higher rates of unmet dental needs [34]. Indeed, studies have shown that parenting stress and supervision improved when economic hardship is reduced [35,36] and, thus, have the potential to minimize if not prevent some ACEs, such as violence in the home and neighborhood.

The early life neighborhood environment is important for favorable health outcomes later in life [37]. Indeed, extant research indicates that neighborhood violence predisposes people to stress that adversely affects their health and wellbeing [38] and residential proximity to recent homicides has been linked to attention and self-regulation processes in very young children [39]. Similarly, neighborhood, family and peer contextual factors were found to be associated with early onset adolescent smoking and alcohol use [40], which are established risk factors for poor oral health outcomes. Our findings were consistent with neighborhood violence being associated with greater odds of poor oral health outcomes.

Although previous studies have found high ACE scores to be associated with poor oral health outcomes without regard to the effect of individual ACEs [21,22,24]. The current study adds to the body of literature with specific focus not only on different oral health outcomes but also on the effect of individual ACE measures. Similar to previous work examining the effect of individual ACEs on oral health [25], our findings suggest that children living in households with financial hardship have greater odds of having poor oral health. Building on previous research, findings from the current study also suggest that after adjusting for child and family characteristics, ACEs related to parental separation and divorce, as well as experiencing caregivers spending time in jail and exposure to domestic violence, were associated with poorer oral health. Additionally, children’s experiences of being treated unfairly due to their race/ethnicity was specifically associated with worse OHRQoL, i.e., reporting the condition of the teeth as fair/poor. Our findings support previous research and suggest that conditions of the family and society impact children’s oral health. Research has shown that children who live “with one or no parents”, children living in poverty and racially minoritized children have higher odds of having unmet dental needs than children living with two parents, children living above the poverty line and white children [41].

### 4.1. Implication of Findings

Pediatric dental providers should be aware of the negative oral health consequences associated with adverse childhood experiences and be trained on best practices, including trauma-informed care when treating children with unique experiences such as exposure to parental divorce, living with mentally ill household members, experiencing violence in the neighborhood or financial hardship. Knowledge and training on these experiences could lead to a better understanding of the patient’s background and a better understanding of how to develop a plan to improve oral health outcomes in the patient. Furthermore, public health officials should work to disseminate educational materials about adverse childhood experiences as well as best dental practices in children to the public, including dental and medical providers, social workers, parents, and other community members. Collaboration among individuals who work or interact with children should be encouraged to better identify and address ACEs’ negative impacts on oral health.

### 4.2. Strengths and Limitations

There are several limitations to the current study. First, the ACE questions in the 2016 NSCH did not include questions related to physical, emotional and sexual abuse nor did it ask about neglect [16]. Indeed, previous studies have reported associations between emotional and sexual abuse and dental fear and anxiety [42]. However, we were unable to replicate this finding in this study. Second, all the measures were self-reported and subject to recall bias and misreporting. Specific to the oral health measures, clinical examinations were not used to determine the actual disease status and, thus, our findings must be interpreted with caution. Moreover, the caregivers answered survey questions pertaining to the children; hence, given the sensitivity of the ACE questions, the potential for inaccurate reporting/misrepresentation of the child’s experiences is possible. Exposure to neighborhood violence had a low prevalence of 3% and may reflect a failure of the children to report an incident to their caregivers. The ACE of violence in the home, with a prevalence of 4%, may be reflective of the caregivers’ refusal to report this information. Third, while all of the oral health measures were self-reported, the reports on OHRQoL were based on a highly subjective question of “How would you describe the condition of (the child’s) teeth?” and not on validated OHRQoL measures such as the Oral Health Impact Profile that assesses the functional, psychological and social domains of OHRQoL [28]. Fourth, there was low representation of non-Hispanic black children in the final sample (*n* = 2,162, 12%), which made stratification according to race/ethnicity challenging. The possibility for reverse causation cannot be ruled out given the likelihood to experience ACEs after the occurrence of a poor oral health outcome. As our study is cross-sectional, we were unable to definitively establish whether the ACEs or the oral health outcomes came first. Risk and protective factors for poor oral health such as diet and oral self-care were not available as adjustment covariates. Lastly, while age may act as a proxy for the length of time exposed to an ACE, all the ACEs measures were limited to exposures in the past 12 months, thus, likely underestimating the actual prevalence of the individual ACE measures and also limiting our ability to assess the chronic nature of ACE exposure on oral health among children and adolescents.

One strength of the current study is that the data were collected on a large, nationally representative sample of children and adolescents. The survey included questions on a wide variety of demographic and health topics; therefore, this allowed for the adjustment of relevant covariates using inverse probability weights that account for selection bias induced by restricting the study to those with complete data [43].

## 5. Conclusions

The findings from this study suggest that exposure to ACEs is associated with poor oral health outcomes in a national sample of U.S. children. These findings are consistent with the previous literature that has shown a relationship between ACEs and negative health outcomes [17] as well as the literature that indicates that experiencing violence or caregiver mental illness is associated with complex health concerns [18]. Indeed, trauma experienced in childhood affects not only health and wellbeing (including poor oral health) in adult life [20], but concurrently affects oral health outcomes during childhood and adolescent years, as our study suggests. One implication of these findings is that the prevention of adverse childhood experiences among young children and adolescents could be effective in preventing specific negative oral health consequences.

This study further supports the evidence that adverse childhood experiences negatively affect the health of children and adolescents and provides a unique perspective on how specific experiences are associated with oral health outcomes. Dental providers, public officials and public health workers should collaborate in making policies and programs geared toward identifying ACEs in children and taking meaningful action to mitigate their harmful oral health consequences.

## Figures and Tables

**Figure 1 ijerph-18-12313-f001:**
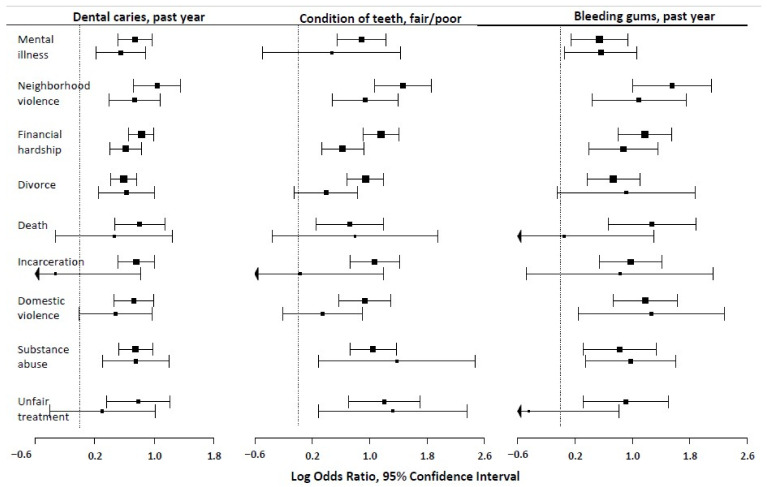
Unadjusted (**top**) and adjusted (**bottom**) associations between exposure to ACEs and poor oral health outcomes among children and adolescent participants of the 2016 National Survey of Children’s Health.

**Table 1 ijerph-18-12313-t001:** Distribution of socio-demographic factors, National Survey of Children’s Health, 2016 (*n* = 41,294).

	Unweighted *n* (Weighted %) or Mean (SE)
**Caregiver characteristics**	
Female gender	28,129 (70.2)
Maternal age at birth, yrs. mean (SE)	29.4 (0.1)
Education	
High School Diploma	5566 (26.4)
Some College, or Associates Degree	12,345 (29.2)
Bachelor’s Degree or Higher	23,383 (44.5)
**Child/ Adolescent characteristics**	
Age, yrs. mean (SE)	8.85 (0.05)
Age group (yrs.)	
0–5	11,458 (30.3)
6–11	12,615 (35.1)
12–17	17,221 (34.6)
Female gender	21,235 (49.0)
Has special healthcare need	9479 (19.7)
Race/ethnicity	
White, Non-Hispanic	29,692 (54.1)
Black, Non-Hispanic	2162 (12.2)
Hispanic	4296 (23.0)
Other, Non-Hispanic	5144 (10.7)
Fair/Poor teeth	1376 (5.2)
Bleeding gums	542 (1.8)
Decayed teeth	3684 (11.2)
Anxiety	3791 (7.4)
Depression	1820 (3.5)
Born in United States	40,137 (96.3)
**Household characteristics**	
Number of children	
1	17,112 (25.2)
2 or 3	22,002 (62.3)
≥4	2180 (12.5)
Language spoken at home	
English	38,843(86.7)
Spanish	1063 (9.0)
Other	1388 (4.3)
Family Structure	
Two parents, currently married	31,410 (68.1)
Two parents, not currently married	2602 (8.6)
One mother, married or not married	4903 (15.5)
Other ^a^	2379 (7.7)
Family Poverty Ratio	
<100%	3453 (18.9)
100–150%	2953 (11.1)
>150%	34,888 (70.0)

^a^ Other comprise the following groups: currently married; other: formerly married; other: never married; other: no parent in household. OHRQoL—Oral Health Related Quality of Life—proxy measure for condition of the child’s teeth and gums.

**Table 2 ijerph-18-12313-t002:** Distribution of Adverse Childhood Experiences, National Survey of Children’s Health 2016.

ACE Measures	Unweighted *n* (Weighted %)
Lived with someone who was mentally ill	3392 (7.9)
Victim of violence/witnessed violence in neighborhood	1250 (3.6)
Hard to cover basics (food/housing)	7942 (25.2)
Parent/guardian divorced/separated	8943 (24.7)
Parent/guardian died	1108 (3.1)
Parent/guardian served time in jail	2272 (7.7)
Parent/guardian were violent toward one another in the home	1843 (5.6)
Lived with someone with a drug/alcohol problem	3605 (9.1)
Treated unfairly because of race/ethnicity	1113 (3.6)

ACE—Adverse Childhood Experiences.

**Table 3 ijerph-18-12313-t003:** Unadjusted and adjusted associations between past year exposure to Adverse Childhood Experiences and past year oral health outcomes, NSCH 2016.

	Decayed Teeth/Cavities		Bleeding Gums ^a^		Fair/Poor Condition of the Teeth	
	**OR (95% CI)**	**AOR ^b^ (95% CI)**	**OR (95% CI)**	**AOR ^b^ (95% CI)**	**OR (95% CI)**	**AOR ^b^ (95% CI)**
Caregiver mental illness	2.10 (1.67, 2.65)	1.73 (1.24, 2.42)	1.71 (1.15, 2.53)	1.75 (1.06, 2.90)	2.43 (1.73, 3.41)	1.60 (0.61, 4.19)
Neighborhood violence	2.82 (2.05, 3.87)	2.09 (1.48, 2.95)	4.72 (2.73, 8.16)	2.97 (1.54, 5.72)	4.32 (2.91, 6.41)	2.56 (1.62, 4.06)
Financial hardship	2.28 (1.93, 2.70)	1.85 (1.50, 2.29)	3.23 (2.21, 4.71)	2.39 (1.48, 3.86)	3.18 (2.48, 4.08)	1.87 (1.40, 2.51)
Caregiver divorce	1.80 (1.52, 2.14)	1.87 (1.28, 2.71)	2.08 (1.44, 3.01)	2.49 (0.95, 6.51)	2.56 (1.99, 3.30)	1.48 (0.95, 2.31)
Caregiver death	2.23 (1.59, 3.14)	1.58 (0.72, 3.47)	3.56 (1.93, 6.55)	1.05 (0.30, 3.66)	2.06 (1.29, 3.30)	2.22 (0.70, 7.09)
Caregiver incarceration	2.13 (1.66, 2.74)	0.72 (0.23, 2.28)	2.64 (1.71, 4.09)	2.28 (0.62, 8.37)	2.92 (2.07, 4.12)	1.03 (0.32, 3.28)
Domestic violence	2.06 (1.57, 2.70)	1.62 (0.99, 2.64)	3.25 (2.08, 5.07)	3.54 (1.28, 9.77)	2.53 (1.76, 3.63)	1.41 (0.81, 2.47)
Drug/alcohol problem in household	2.11 (1.68, 2.66)	2.11 (1.35, 3.31)	2.28 (1.37, 3.79)	2.66 (1.42, 5.00)	2.85 (2.07, 3.94)	3.98 (1.33, 11.8)
Treated unfairly due to race/ethnicity	2.19 (1.43, 3.37)	1.35 (0.66, 2.73)	2.47 (1.37, 4.47)	0.64 (0.18, 2.21)	3.34 (2.04, 5.48)	3.77 (1.34, 10.6)

^a^ In the past 12 months. ^b^ Adjusted for child age, child sex, race/ethnicity, primary language, family structure, primary adult sex, child born in USA, child anxiety, child depression, primary adult education, family poverty ratio, mother’s age at birth, number of children in household, special healthcare needs status. OHRQoL—Oral Health Related Quality of Life, proxy measure for condition of the teeth and gums; defined by the description of child’s teeth. OR- Odds Ratio; CI-Confidence Interval; AOR-Adjusted Odds Ratio.

## Data Availability

Data are publicly available on the NSCH website at https://www.childhealthdata.org/learn-about-the-nsch/NSCH (28 October 2021).

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
