# Peer review of "Adverse Childhood Experiences and Oral Health Outcomes in U.S. Children and Adolescents: A Cross-Sectional Study of the 2016 National Survey of Children’s Health"

_ijerph, 2021, doi:10.3390/ijerph182312313_

Round 1

Reviewer 1 Report

The aim of this study was to investigate oral health outcome and oral health related quality of life of a national sample of US children and adolescents. There are some issues that authors need to address for possible consideration for publication in this journal, as detailed below:

  • Abstract: should avoid abbreviations. however, if the authors choose to use them, the definition should be placed, for example, ACE in line 13.
  • Introduction: authors should briefly highlight the impact of the studied outcomes (dental caries, bleeding gums) and quality of life, since there are already numerous studies published in this topic.
  • Objective: There are numerous tools validated in the literature to measure the OHRQoL. However, in this study, none of these were used. For this reason, the presented objective must be reformulated.
  • Materials and Methods: how was the sample selection made? Inclusion and exclusion criteria are missing. The authors must carry out the characterization of the collected sample. Did the authors take any precautions to avoid bias? As mentioned above, there are validated tools to measure OHRQoL that are not used in this study. I recommend the reformulation of the methods of line 87 to 91. Is the questionnaire used properly validated? How many responses were obtained in each ACE? Were the number of responses homogeneous? How long was the survey available? should describe the statistical software used.
  • Conclusion: should provide readers with a brief summary of the main conclusions and not a comparison with the literature (line 301-306). The conclusion must be rephrased.

Author Response

Response to Reviewer 1 Comments

The authors wish to thank the reviewer for the time spent reviewing our manuscript and for their constructive feedback.

Reviewer 1

The aim of this study was to investigate oral health outcome and oral health related quality of life of a national sample of US children and adolescents. There are some issues that authors need to address for possible consideration for publication in this journal, as detailed below:

Abstract: should avoid abbreviations. however, if the authors choose to use them, the definition should be placed, for example, ACE in line 13.

Response: All abbreviations in the abstract have been defined when they first appeared.

Introduction: authors should briefly highlight the impact of the studied outcomes (dental caries, bleeding gums) and quality of life, since there are already numerous studies published in this topic.

Response: additional information has now been included on the impacts of dental caries in general and specific to quality of life.

Objective: There are numerous tools validated in the literature to measure the OHRQoL. However, in this study, none of these were used. For this reason, the presented objective must be reformulated.

Response: We used self-rated condition of the teeth as a proxy for OHRQoL because the OHRQoL instrument was not used in this data source and so we do not have those data. The objective has been updated to reflect this.

Materials and Methods: how was the sample selection made? Inclusion and exclusion criteria are missing. The authors must carry out the characterization of the collected sample. Did the authors take any precautions to avoid bias? As mentioned above, there are validated tools to measure OHRQoL that are not used in this study. I recommend the reformulation of the methods of line 87 to 91. Is the questionnaire used properly validated? How many responses were obtained in each ACE? Were the number of responses homogeneous? How long was the survey available? should describe the statistical software used.

Response: All of the points raised by the reviewer are important and were addressed in the manuscript. The specific locations have been highlighted in yellow for the reviewer’s convenience. These are publically available national dataset and we, the investigators were not involved in the data collection process. We addressed the limitations of the ACEs measures that we have in the limitations section. The software we used was SAS v. 9.4

Conclusion: should provide readers with a brief summary of the main conclusions and not a comparison with the literature (line 301-306). The conclusion must be rephrased.

Response: The first and third sentences of the conclusion summarized our main findings. These are now highlighted in yellow.

Reviewer 2 Report

Very interesting work and of great relevance for public health. Here are some comments:

Introduction

-The introduction is well structured, however it is important to go deeper into the adverse experiences in childhood: the classification (if any), and what are the adverse experiences in childhood (only mention some)? You talk about nine ACEs but never say which they are. In general more weight should be given in the introduction, as it is the foundation of this article.

-Quality of life related to oral health is considered in the analysis, but is not addressed in the introduction.

Line 13: The abbreviation “ACEs” has not been previously defined in the abstract.

Line 19: “(0.1)” Is this the standard deviation?

Materials and Methods

Line 97-100: This limitation of not having data from a validated instrument to measure oral health related quality of life (OHRQoL), should be widely taken up in the discussion.

Line 109: The age of the caregiver should also be considered, it is not the same to be in the care of an adolescent, than of an adult or older adult.

Results

-Tables must be after the text where they were referred. Review the guide for authors.

-All the tables must contain the meaning of the abbreviations at the bottom of the same, in such a way that the reader can understand them without necessarily going to the text.

-In the methods they say that the quality of life related to oral health will be analyzed through a proxy variable, but in the results they do not present anything in this regard.

Line 141: Please report the minimum and maximum of the age variable.

Table 1: It is difficult to read the values in the table as percentages and standard deviations are presented in the same way.

Line 185: The information in the appendices is to provide additional information and should not be referenced in the main text of the manuscript.

Table 3: The table format does not allow adequate visualization of the confidence intervals.

Discussion

-Please mention your proposals for future studies.

-Among the limitations, it is important to highlight that the information on the oral health status does not come from a clinical examination and the precautions that must be considered in this regard when interpreting the information resulting from the researchers' analysis.

Author Response

Response to Reviewer 2 Comments

The authors wish to thank the reviewer for the time spent reviewing our manuscript and for their constructive feedback.

Very interesting work and of great relevance for public health. Here are some comments:

Introduction

-The introduction is well structured, however it is important to go deeper into the adverse experiences in childhood: the classification (if any), and what are the adverse experiences in childhood (only mention some)? You talk about nine ACEs but never say which they are. In general more weight should be given in the introduction, as it is the foundation of this article.

Response: We have now listed in the introduction section the 9 ACEs we looked at in this study.

-Quality of life related to oral health is considered in the analysis, but is not addressed in the introduction.

Response: We have added information on oral health related quality of life to the introduction section and wish to reiterate that the measure we had was how the caregivers rated the condition of the teeth and gums of the child. We used this measure as a proxy for oral health related quality of life.

Line 13: The abbreviation “ACEs” has not been previously defined in the abstract.

Response: We have now defined ACE the first time it appears in the abstract.

Line 19: “(0.1)” Is this the standard deviation?

Response: This is the standard error. We have it in parenthesis as (SE)

Materials and Methods

Line 97-100: This limitation of not having data from a validated instrument to measure oral health related quality of life (OHRQoL), should be widely taken up in the discussion.

Response: Again we do not have measures on OHRQoL, we used the condition of the teeth as a proxy measure. We have now added a sentence to the limitations section on not using a validated OHRQoL measure.

Line 109: The age of the caregiver should also be considered, it is not the same to be in the care of an adolescent, than of an adult or older adult.

 Response: The data source only has information on the age of the mother when the study child/adolescent was born.

Results

-Tables must be after the text where they were referred. Review the guide for authors.

Response: Thank you

-All the tables must contain the meaning of the abbreviations at the bottom of the same, in such a way that the reader can understand them without necessarily going to the text.

Response: Thank you. Tables have been formatted to be standalone

-In the methods they say that the quality of life related to oral health will be analyzed through a proxy variable, but in the results they do not present anything in this regard.

 Response: We again wish to reiterate that we do not have actual OHRQoL measure. What we had was condition of the teeth/gums which we used as a proxy for OHRQoL. We have now added to the limitations section, the impacts of not having the actual OHRQoL measure.

Line 141: Please report the minimum and maximum of the age variable.

 Response: We are not sure what the reviewer is referring to here. Age of the child/adolescents or maternal age at birth?

Table 1: It is difficult to read the values in the table as percentages and standard deviations are presented in the same way.

Response: we have made a note which ones are mean and SE

Line 185: The information in the appendices is to provide additional information and should not be referenced in the main text of the manuscript.

 Response: we respectfully disagree with the reviewer on this point

Table 3: The table format does not allow adequate visualization of the confidence intervals.

Response: Table 3 was formatted as landscape and should allow the CI be visualized in its entirety 

Discussion

-Please mention your proposals for future studies.

Response: Thank you for the comment. We do not currently have a proposal for future studies

-Among the limitations, it is important to highlight that the information on the oral health status does not come from a clinical examination and the precautions that must be considered in this regard when interpreting the information resulting from the researchers' analysis.

Response: We discussed in the limitations that the oral health measures are self-reported and have now added a sentence to the effect of the reviewer’s point.

Round 2

Reviewer 1 Report

All recommendations were considered by the authors. Congrats!